# Self-Efficacy and Mental Health Help-Seeking Behavior of World Trade Center Health Registry Enrollees, 2015–2016

**DOI:** 10.3390/ijerph19127113

**Published:** 2022-06-10

**Authors:** Sascha K. Garrey, Erin Takemoto, Lysa Petrsoric, Lisa M. Gargano

**Affiliations:** World Trade Center Health Registry, New York City Department of Health and Mental Hygiene, New York, NY 11101, USA; etakemoto@health.nyc.gov (E.T.); lsilverstein@health.nyc.gov (L.P.); lisa.gargano@health.ri.gov (L.M.G.)

**Keywords:** self-efficacy, mental health, help-seeking, September 11th, disaster epidemiology, PTSD, unmet mental healthcare need

## Abstract

The September 11th World Trade Center (WTC) disaster resulted in an elevated prevalence of Post-Traumatic Stress Disorder (PTSD) among those directly exposed, yet lower than expected rates of mental health treatment seeking and high levels of reported perceived unmet mental healthcare need were observed in this population in the years following. Self-efficacy, an individual’s self-perception of their ability to succeed in specific situations or accomplish a task or goal, may in part explain this discrepancy; however, little is known about its interplay with the help-seeking behaviors of disaster-exposed populations. We used WTC Health Registry data (*n* = 11,851) to describe the relationship between self-efficacy and three outcomes related to help-seeking behavior: (1) seeking mental health treatment, (2) perceived unmet mental health care needs, and (3) satisfaction with mental health treatment. Multinomial logistic regression models were used to estimate adjusted odds ratios (AORs) and 95% confidence intervals (CI). We found a dose-response relationship between self-efficacy score and mental health help-seeking: for every one unit increase in self-efficacy score, we observed a 6% increase in the odds of having treatment 4 to 12 months ago (OR = 1.06, CI: 1.03–1.09), a 7% increase in the odds of having had treatment 1 to 2 years ago (OR = 1.07, CI: 1.04, 1.09), and a 10% increase in the odds of having sought treatment 2 or more years ago (OR = 1.10, CI: 1.08, 1.12) compared to those who had sought treatment more recently. An understanding of individual self-efficacy may help improve post-disaster mental health treatment in order to provide more tailored and helpful care.

## 1. Introduction

The long-term impacts of the 11 September 2001 (9/11) terrorist attacks on the World Trade Center (WTC) are still felt to this day, in both the physical and mental health of those directly exposed to the disaster. Post-traumatic stress disorder (PTSD) was one of the most common mental health morbidities resulting from the attacks [1] and continues to be a significant health burden amongst those directly exposed. Depending on the population studied, the prevalence of 9/11-related PTSD ranges from 3.8% to 29.6% [2]. However, despite the high prevalence of PTSD and other serious mental health conditions, one study found that in the year following the attacks, rates of those seeking mental health services were lower than expected [3]. Additionally, perceived unmet mental healthcare need, a construct that measures an individual’s subjective assessment as to whether they received the care that they needed in a given time period [4], is a persistent issue among 9/11 survivors [5]. One study found that 20% of those diagnosed with post-9/11 mental health conditions reported having an unmet mental healthcare need five to six years after the disaster [6]. This study also found that those with PTSD were more likely to seek mental health treatment compared to those without PTSD, but they were also more likely to report an unmet mental healthcare need [6]. Unmet mental healthcare need was reported to be as high as 31% among a population directly exposed to 9/11, irrespective of mental health diagnosis, 15 years after the disaster, but those with the greatest exposure to the disaster reported less delay in seeking treatment [7].

There are several possible explanations for the lower than expected levels of mental health care treatment and high levels of unmet mental healthcare need among trauma-exposed populations, including individual-level differences in help-seeking tendencies. Self-efficacy, or an individual’s self-perception of their ability to succeed in specific situations or accomplish a task or goal [8], is one such individual-level characteristic. The construct of self-efficacy is rooted in social cognitive theory which views human beings as fundamentally autonomous agents who possess capabilities of self-appraisal and self-reflection, which they can proactively apply to their own self-development and adaption during times of upset or change [8]. In this model, action or behavior is inextricable to the beliefs one has about their own efficacy in that it is thought to be the outward manifestation of such beliefs [9]. Help-seeking is a potential behavioral means of achieving the goal of better mental health. It fits into the framework of social cognitive theory in that the behavior of seeking help for physical or mental health symptoms requires (1) a subjective appraisal of oneself as possessing a health problem that is beyond one’s own abilities to address and (2) an internal belief in one’s own ability to benefit from the sought help. 

Although the relationship between self-efficacy and mental health help-seeking behavior is supported theoretically, the scientific evidence is mixed. Judd et al. [10] found that self-efficacy among residents of rural areas in Australia was inversely related to help-seeking: lower self-efficacy was associated with a higher likelihood of seeking professional help. However, this study was limited by its relatively small sample size (*n* = 822), and it examined older individuals from small, rural villages, thus constricting the applicability of findings to urban or more general populations. In contrast, Andersson et al. [11] studied a sample of the general Swedish population and found that low self-efficacy was associated with a greater likelihood of reporting having mental illness and a lower likelihood of seeking mental health treatment compared to high self-efficacious individuals. These results expand those of Judd et al. in that they were derived from a general population sample composed of both rural and urban dwellers; however, they may also reflect behavioral norms that are specific to the social and cultural environment of Sweden. Further research is warranted to examine the construct in the context of help-seeking behaviors in different countries. Self-efficacy is also underrepresented as a central study consideration in the mental health help-seeking literature. In review of 350 studies on mental health help-seeking, Jackson et al. [12] found that only one [10] study considered the role that self-efficacy may play in help-seeking, and concluded that additional research is needed to determine the nature of the relationship. Further, the association between self-efficacy and help-seeking behavior may be different among disaster-exposed populations compared to the more general populations that have been studied. 

Self-efficacy has a documented positive association with post-traumatic recovery [13]. It is theorized that those with higher levels of self-efficacy who are subject to traumatizing events have a greater ability to employ more proactive coping strategies and to be less likely to submit to distorted, self-deprecating thought patterns [13]. Although self-efficacy has been studied readily in populations exposed to natural disasters [13,14,15], little is known about the role self-efficacy plays in post-traumatic recovery in populations exposed to mass terrorist attacks, who have been shown to exhibit higher prevalence of PTSD and depression than the general population. Self-efficacy is a construct that is thought to be relatively malleable, varies in response to individual learning experiences, and can be improved upon therapeutically [16]. For populations affected by mass disasters, including mental health interventions that target the bolstering of individual self-efficacy beliefs may be an important consideration in disaster preparedness planning. Thus, additional studies are warranted to further elucidate the association between self-efficacy and mental health help-seeking behavior.

To examine the association of self-efficacy and mental health help-seeking behaviors among persons in a post-disaster context, we used data from the World Trade Center Health Registry (WTCHR), a longitudinal cohort study of 71,426 individuals who were directly exposed to the terrorist attacks on 9/11 and who exhibited a higher prevalence of PTSD than the general US population [17]. The goals of this cross-sectional study were to use a population sample collectively exposed to a mass disaster to describe the relationship between self-efficacy and three independent mental health help-seeking outcomes: (1) seeking mental health treatment, (2) perceived unmet mental health care needs, and (3) satisfaction with mental health treatment among those who received recent treatment. 

## 2. Materials and Methods

### 2.1. Study Population

The World Trade Center Health Registry (WTCHR) is a longitudinal closed cohort study of 71,426 individuals directly exposed to the 9/11 disaster. Registry enrollees include adults and children who belong to one or more of five study eligibility groups: rescue/recovery workers and volunteers, lower Manhattan residents, area workers, school students and staff, and passersby. The WTCHR has completed five surveys to date: Wave 1 (W1, 2003 to 2004), Wave 2 (W2, 2006 to 2007), Wave 3 (W3, 2011 to 2012), Wave 4 (W4, 2015 to 2016), and Wave 5 (W5, 2020 to 2021). Further details on WTCHR recruitment, enrollment, and survey administration can be found elsewhere [18]. The institutional review boards at the Centers for Disease Control and Prevention and the New York City Department of Health and Mental Hygiene approved the WTCHR protocol.

### 2.2. Study Measures

All outcome measures were taken from the W4 survey.

Outcome 1: Mental Health Help-Seeking.

A four-level mental health help-seeking variable was created to capture an enrollee’s most recent counselling or therapy visit. Enrollees responded to the question “When was the most recent time you received counseling or therapy?” using one of the following response options: (1) less than four months ago, (2) four months to less than one year ago, (3) one to two years ago, and (4) greater than two years ago.

Outcome 2: Unmet Mental Health Care Need.

A dichotomous (yes/no) unmet mental health care need variable was derived from two questions. Enrollees who responded (1) that they did not need mental health care in the last 12 months or (2) needed mental health care and received that care were classified as not having an unmet mental health care need. Enrollees who reported that they needed mental health care in the last 12 months and did not receive such care were classified as having an unmet mental health care need.

Outcome 3: Perception of the Helpfulness of Most Recent Counselling or Therapy Session.

Among the subset of enrollees who reported having received mental health counselling or treatment in the last 12 months, we examined their perception of how helpful the most recent counselling or therapy visit, based on a four-level satisfaction variable: (1) not at all helpful, (2) slightly helpful, (3) somewhat helpful, and (4) very helpful.

Exposure: Self-Efficacy.

Self-efficacy was measured using an abridged version of the General Self-Efficacy scale (GES) [19], a twelve-item scale used to assess an individual’s level of perceived self-efficacy. Five statements of the total 12 self-efficacy statements of the GES were queried on the W4 survey. These five statements had good internal consistency and yielded an alpha of 0.80. GES responses were scored between 1 (not at all true) and 4 (exactly true) and an aggregate score was summed, with the continuous score ranging between 5 and 20. Higher scores indicate higher self-efficacy.

Confounders: Age, Education, Income, Social Support, Social Integration, Depressive Symptoms.

For all three outcomes, the following potentially confounding variables were selected a priori. Wave 4 demographic variables included age (<40 years, 41–64 years, ≥65 years), education (high school/GED or less, at least some college or technical degree, Bachelor’s degree, postgraduate degree), and household income (<$74,999, $75,000–$99,999, ≥$100,000). Sex (male, female) was documented at W1. We also included a measure of social support/integration, probable PTSD, and depressive symptoms, all measured at W4.

Social support status was measured via an abridged version of the Medical Outcomes Study Social Support Survey, which uses five items to assess five different domains of functional social support, or the degree to which interpersonal relationships serve particular functions in one’s daily life [20]. Enrollees were asked to rank four items on different domains of functional social support (0 = none of the time to 4 = most of the time). In our analysis, continuous scores for social support status ranged from 0 to 20 [5].

Social integration was measured using Cohen et al.’s [21] social integration index which assesses an individual’s participation in a range of social relationships/networks. Enrollees were asked to indicate using a yes or no response whether certain social integration factors were experienced in the last 30 days. Reporting one or more close friends at the time of the survey was also considered a means of social integration. A final count variable that ranged from 0 to 4 represented the number of social integration sources (0 = none to 4 = four or more sources) [22].

Probable PTSD was measured using the PTSD Checklist (PCL-S) at W4. The PCL-S is a self-reported 17-symptom scale corresponding to DSM-IV criteria and asks about symptoms in relation to a “stressful experience,” specifically the WTC disaster. Respondents were asked how much each of 17 symptoms bothered them over the last thirty days, ranging from 1 = not at all to 5 = extremely, and the scores from the 17 items were summed. Based on their aggregated PCL scores, enrollees with PCL scores of 44 or higher were considered to have probable 9/11-related PTSD (hereafter, PTSD) [23,24].

Depressive symptoms at W4 were assessed using enrollee responses to eight items on the Patient Health Questionnaire Version 8 (PHQ-8), a validated self-reported instrument that contains eight of the nine DSM-IV depression diagnostic criteria [25]. Responses were rated on a five-point Likert scale (0 = not at all to 4 = nearly every day). PHQ-8 items on the W4 survey had a high internal consistency (α = 0.92). PHQ-8 measures were included as continuous scores that ranged from 0 to 24.

### 2.3. Analytic Sample 

For inclusion in the current study, enrollees had to have completed both the W1 and W4 surveys (*n* = 36,862) and they had to report that they had at least one session of counseling or therapy after 9/11 (*n* = 13,926). Those who reported that their most recent post-9/11 counseling or therapy session was over two years ago (*n* = 214) and those with missing information on their most recent counseling or therapy visit (*n* = 1861) were excluded from the sample. The final analytic sample size for outcome 1 (mental health help-seeking) and 2 (perceived unmet mental healthcare need) was 11,851. For our analysis of outcome 3 (perception of helpfulness of the most recent counseling or therapy session), we restricted our study sample to those who reported receiving counselling or therapy within the last year at W4 as the survey question asking about enrollee’s satisfaction with their mental health treatment was only posed to this cohort of individuals (*n* = 5245).

### 2.4. Data Analysis

We used Chi square tests for dichotomous variables and bivariate multinomial logistic regression to tests for continuous variables to assess their association with the outcomes.

We used multinomial logistic regression to model the relationship between mental health help-seeking (outcome 1) and self-efficacy. Adjusted odds ratios (AOR) and 95% confidence intervals (CI) were computed using those receiving treatment less than four months ago as referent. We used logistic regression to model the relationship between perceived unmet mental healthcare need (outcome 2) and self-efficacy. AORs and 95% CIs were computed with those who had no perceived unmet mental healthcare need as referent. For perception of helpfulness of treatment (outcome 3), we used multinomial logistic regression to model the association with self-efficacy. ORs and 95% CIs were computed with those finding treatment “not at all helpful” as referent. 

All models controlled for all confounding variables (age, education, income, social support, social integration, depressive symptoms) and all analyses were performed in SAS Enterprise Guide (Cary, NC, USA), version 9.4.

## 3. Results

A majority of the study sample was male (53.8%), aged 41 to 64 years (71.1%), reported having a Bachelor’s degree or higher (62.7%), and had a household income of $100,000 or higher (51.3%) (Table 1). The mean self-efficacy score for the entire study sample was 15.4 (SD: 3.2). Men ≥ 65 years of age with a postgraduate education were the most represented in the sample group that had sought mental health treatment less than four months ago (Table 2). Those with the largest perceived unmet mental healthcare need were men, aged 65 and older, who reported having some college education or high school or less (Table 3). Women aged 65 years or above with a postgraduate degree were the most likely to perceive their mental health treatment as “very helpful”, whereas men aged 18 to 40 with a high school diploma, GED, or less were most represented among those who reported that their mental health treatment was “not helpful at all”.

### 3.1. Outcome 1: Mental Health Help-Seeking

Over one-third of respondents reported that they had sought mental health counselling or therapy less than four months ago, and 43.2% of respondents reported that their most recent counselling or therapy session was over two years ago (Table 2). Mean self-efficacy score increased as time since last counseling or therapy session increased. For instance, the mean self-efficacy score was 14.5 (SD = 3.5) for those who had sought mental health treatment less than four months ago, whereas it was 16.1 (SD = 2.8) for those who had sought mental health treatment over two years ago (*p* < 0.0001).

In the adjusted multinomial model (Table 4), we observed that a higher self-efficacy score was associated with a longer time since last mental health treatment. For every one unit increase in self-efficacy score, we observed a 6% increase in the likelihood of having treatment four to 12 months ago (OR = 1.06, CI: 1.03–1.09), a 7% increase in the likelihood of having had treatment one to two years ago (OR = 1.07, CI: 1.04, 1.09), and a 10% increase in the likelihood of having sought treatment two or more years ago (OR = 1.10, CI: 1.08, 1.12) compared to those who had sought treatment within the last four months. To test for trend, we ran a Chocran-Mantel-Haenszel test, which yielded a General Association *p*-Value of < 0.0001, indicating a dose-response relationship between time since last counselling or therapy session and self-efficacy score.

### 3.2. Outcome 2: Unmet Mental Health Care Need

Having an unmet mental health care need at W4 was reported by 12.3% of the study sample. Those who reported having an unmet mental healthcare need had a mean self-efficacy score of 13.9 (SD = 3.3; *p* < 0.0001), whereas those who did not report having an unmet mental healthcare need had a mean self-efficacy score of 15.6 (SD = 3.1; *p* < 0.0001) (Table 3). 

In the adjusted model, we observed that there was no relationship between reporting an unmet mental health care need and self-efficacy score. (Table 4)

### 3.3. Outcome 3: Perception of Helpfulness of Mental Health Treatment

The mean self-efficacy score of the sub-sample who reported receiving treatment in the last year was 14.6 and the majority (79.7%) reported that they found their most recent session of mental health treatment to be “very helpful,” or “somewhat helpful.” Those who reported finding their most recent mental health treatment to be very helpful had the highest mean self-efficacy score (15.5), whereas those who found their most recent mental health treatment to be not helpful at all had the lowest mean self-efficacy score (12.9; *p* < 0.0001) (Table 5).

In the adjusted model, for each unit increase in self-efficacy score, we observed a 6.8% decrease in the odds of reporting most recent mental health treatment to be somewhat helpful (OR = 0.932; CI: 0.907, 0.958) and an 8.5% decrease in the odds of reporting finding their most recent mental health treatment to be slightly helpful (OR = 0.915; CI: 0.884, 0.947) compared to those who found treatment to be very helpful. For those who reported their most recent mental health treatment to be not at all helpful, there was no difference in self-efficacy score compared to the referent group (Table 4).

## 4. Discussion

This study is one of a growing few to examine self-efficacy in the context of mental health help-seeking behavior, and perceived unmet mental healthcare need in a large, post-disaster population. We found that those with lower self-efficacy scores were more likely to report having been to counselling or therapy more recently compared to those with higher self-efficacy scores, and those with lower self-efficacy were also more likely to report having an unmet mental healthcare need. For those who had received mental health treatment within the last 12 months, the likelihood of having a more positive view of the utility of that treatment increased as self-efficacy score increased. As self-efficacy is a malleable construct that can be improved upon therapeutically and because it has been shown to bolster post-traumatic recovery [26], these findings have important considerations for self-efficacy in disaster preparedness and post-disaster mental health intervention planning.

We observed a clear dose-response relationship between mental health help-seeking and self-efficacy scores: the level of self-efficacy increased as the reported time since an enrollee’s last counseling or therapy session grew. Those with higher self-efficacy may have felt more empowered to take care of their mental health problems on their own and thus did not seek care as readily as low self-efficacy individuals. These individuals’ beliefs about their ability to cope with problems or achieve goals play a unique role in their decision whether to seek mental health treatment, a circumstance which, in the long term, could be beneficial or problematic for mental health outcomes. Higher self-efficacy individuals may draw from other external resources to improve or maintain their mental health status such as through seeking social support and consequently go longer without care while also maintaining their mental health status [27]. However, it is unknown if those with higher self-efficacy can adequately resolve mental health problems on their own, which could contribute to worse long-term mental health over time. Kessels and Steinmayr [28] found that male high school students who exhibited more self-reliant or traditionally masculine traits tended have poorer educational outcomes than their female counterparts and that this association was partially explained by negative attitudes towards help-seeking. In a sample of 250 young-adult Rwandans suffering from depression and/or suicidal ideation, Umubyeyi [29] found that higher self-efficacy was associated with higher confidence in one’s own personal strength to cope with their mental health problems on their own. The findings from these studies support the theory that having higher perceived self-efficacy may keep some from seeking help, irrespective of truly needing that help. We are unable to determine in our study if the relationship between self-efficacy and a longer time since treatment is indicative of overall better or worse long-term mental health outcomes. Future studies are warranted to elucidate whether individuals with high self-efficacy that need mental health treatment are adequately able to resolve their own needs without seeking treatment or if their perceptions of their ability to individually cope are ultimately a detriment to their mental health.

Our finding that a shorter time since last mental health treatment was observed among those with lower self-efficacy may also be explained by a difference in the burden of mental health conditions. It is possible that those with lower self-efficacy have a greater burden of mental health conditions and are, consequently, in greater need of mental health treatment compared to those with higher self-efficacy. We did control for the most prevalent mental health conditions in our population, depression and PTSD, and still observed a significant association between self-efficacy and time since last mental health visit. This does not rule out the possibility that those with lower self-efficacy also have more severe mental health conditions, necessitating more recent treatment, as disease severity was not available in our data. Incorporating disease severity in addition to disease burden in future studies will further improve the understanding of the relationship between self-efficacy and seeking mental health treatment.

A small proportion of our sample reported having an unmet mental health care need at W4; however, those who reported having an unmet mental health care need had a lower mean self-efficacy score compared to those who did not report having an unmet mental health care need. In social cognitive theory, whether a person appraises themselves and their lives optimistically or pessimistically is a function of self-efficacy [8]. The optimism or pessimism that results from a person’s self-efficacious constitution plays a role in the shaping of that person’s outcome expectations and their perception of their ability to achieve the goals that they have set. In this study, we observed that lower self-efficacy individuals were more likely to recently seek mental health treatment compared to high self-efficacy individuals; however, their perception that their need for care had been “met” may have been appraised pessimistically. Perhaps those individuals in our sample with lower self-efficacy exhibited a negative bias towards how they perceived their ability to heal psychologically and that their expectation of having their mental health care needs “met” is impeded by a pessimistic appraisal of themselves. Thus, it may mean, to some extent, that reporting an unmet mental healthcare need was a function of self-perception, more so than a true unmet need; however, we are not able to determine this with any certainty given our study constraints. Further research needs to be pursued to look into the reasons for which those with low and high self-efficacy perceived their need for mental health care as being met or unmet.

We found that among individuals who had sought counselling or therapy in the last 12 months, those with lower self-efficacy were more likely to view that mental health treatment as unhelpful compared to higher self-efficacy individuals. It may be that those with lower self-efficacy needed more specialized mental health treatment, thus standard counselling or therapy would not be seen as “helpful” as it may not have been framed in a way that is specific to bolstering one’s belief in their agency. In a study of a group of survivors of the 1995 Oklahoma City bombing, Benight et al. [26] found that the judgements of coping self-efficacy were the most important explanatory factor in the differences in post-traumatic recovery. These findings suggest that a person’s self-perception of their coping capability and their perception of their ability to persevere through their trauma may be more informative from a treatment standpoint than relying primarily on the symptoms of distress. For instance, it could be that for the group of low self-efficacy individuals, counselling is not seen as “helpful” as the counselling they received may have primarily focused on the more obvious manifestation of their distress (i.e. the depression or PTSD symptoms themselves) as opposed to their perception of their distress. However, we did not have detailed information on the source of the unmet need or the reason for which individuals found their mental health treatment to be unhelpful (e.g., lack of financial resources, quality of the care). In the context of care in the aftermath of mass disasters, these findings highlight the need for individualized mental health treatment that considers not only a person’s PTSD symptoms, but also their actual ability to cope with those symptoms given their level of perceived self-efficacy. Also, “time since care” may be an inadequate metric to focus on in isolation in help-seeking studies and that the greater context of a person’s self-perception ought to be taken into consideration when possible. 

This study is subject to several limitations. Firstly, the scale used to measure our central exposure variable, self-efficacy, was derived from a generally worded scale. According to Bandura [30], measures for self-efficacy are usually centered on a specific task or problem-specific judgments and future studies should consider measures that are worded more specifically around the domain of help-seeking. Secondly, due to our cross-sectional study design, we are not able to clearly define the direction of the association between self-efficacy, mental health help-seeking, unmet mental healthcare, and the perception of helpfulness of counseling or therapy. Thirdly, although there were other waves of data available to our study team, we opted not to include them in the present study because our exposure variable, self-efficacy score, did not appear on the WTCHR survey waves until W4. Finally, because we studied this association in a population exposed to the events of 9/11, generalizations of our findings to a non-trauma exposed population may be limited. Despite these limitations, we were able to expand on the existing help-seeking and self-efficacy literature by considering social cognitive theory in a large population sample and to examine self-efficacy in the context of a post-disaster population. Further, the effects we found were robust and mostly statistically significant, indicating that the consideration of self-efficacy in help-seeking research in populations exposed to mass disasters is an important area for further research.

## 5. Conclusions

In sum, findings in this study indicated that, in a post-disaster population, those with lower-self efficacy were more likely to receive mental health treatment more recently than higher-self efficacy individuals; however, they were also more likely to report an unmet mental healthcare need and less likely to view their mental health treatment as helpful. These findings suggest that although those with low-self efficacy tended to have more negative views of their ability to exact change in their lives, they proactively sought help more often for their mental health problems and that more specialized mental health treatment for those with low self-efficacy may be warranted. An understanding of individual self-efficacy may help improve post-disaster mental health treatment in order to provide more tailored and helpful care.

## Figures and Tables

**Table 1 ijerph-19-07113-t001:** Totals for Sociodemographic and Mental Health Characteristics of World Trade Center Health Registry Enrollees who had at least one session of counselling or therapy after 9/11, 2015–2016 (*n* = 11,851).

Variable	Total *n* (%) *
Time of Most Recent Counselling or Therapy Visit (Outcome 1)	
<4 mths ago	4147 (35.0)
4 mths 0 < 1 year ago	1098 (9.3)
1–2 years ago	1491 (12.6)
2+ years ago	5115 (43.2)
Perceived Unmet Mental Healthcare Need (Outcome 2)	
Yes	1455 (12.3)
No	10,369 (87.7)
Perception of helpfulness of most recent counselling or therapy visit (Outocome 3)	
Very helpful	2145 (41.6)
Somewhat helpful	1969 (38.1)
Slightly helpful	846 (16.4)
Not at all helpful	202 (3.9)
General Self Efficacy Score	
Mean	15.4
SD	3.2
Sex	
Male	6380 (53.8)
Female	5471 (46.2)
Age; M = 3	
18–40 years	1596 (13.5)
41–64 years	8421 (71.1)
65+ years	1831 (15.5)
Education; M = 88	
Highschool/GED or less	1254 (10.7)
At least some college or technical degree	3138 (26.7)
Bachelor’s Degree	3620 (30.8)
Post Graduate Degree	3751 (31.9)
Household Income; M = 519	
<$74,999	3990 (35.2)
$75,000–$99,999	1532 (13.5)
≥$100,000	5810 (51.3)
PCL ≥ 44; M = 584	
Yes	2839 (25.2)
No	8428 (74.8)
Sources of Social Integration; M = 221	
0	117 (1.0)
1	589 (5.1)
2	5221 (44.9)
3	3936 (33.8)
4	1767 (15.2)
Social Support	
Mean	13.1
SD	5.4
PHQ-8 Score	
Mean	7.0
SD	6.2

Abbreviation: N: number; M: missing; GED: General Equivalency Degree; SD: standard deviation; PCL: PTSD Checklist; PHQ-8: Personal Health Questionnaire Depression Scale 8th Edition. * Column percentages.

**Table 2 ijerph-19-07113-t002:** Sociodemographic and Mental Health Characteristics of World Trade Center Health Registry Enrollees Who Had at Least One Session of Counselling or Therapy After 9/11, Stratified by Time of Most Recent Counselling or Therapy Visit (i.e., Outcome #1), 2015 to 2016 (*n* = 11,851).

	Time of Most Recent Counselling or Therapy Visit (Outcome 1)				
Variable	<4 mths ago	4 mths 0 < 1 year ago	1–2 years ago	2+ years ago	*p*-Value
	*n* (%) *	*n* (%) *	*n* (%) *	*n* (%) *	
Totals	4147 (35.0)	1098 (9.3)	1491 (12.6)	5115 (43.2)	n/a
General Self Efficacy Score					
Mean	14.5	15.2	15.5	16.1	<0.0001 **
SD	3.5	3.2	3.0	2.8	
Sex					
Male	2258 (35.4)	582 (9.1)	826 (12.8)	2724 (42.7)	0.5511 ***
Female	1889 (34.5)	516 (9.4)	675 (12.3)	2391 (43.7)	
Age; M = 3					
18–40 years	546 (34.2)	164 (10.3)	208 (13.0)	678 (42.5)	0.7329 ***
41–64 years	2950 (35.0)	770 (9.1)	1064 (12.6)	3637 (43.2)	
65+ years	650 (35.5)	164 (9.0)	219 (12.0)	798 (43.6)	
Education; M = 88					
Highschool/GED or less	499 (39.8)	128 (10.2)	143 (11.4)	484 (38.6)	<0.0001 ***
At least some college or technical degree	1068 (34.0)	311 (9.9)	427 (13.6)	1332 (42.45)	
Bachelor’s Degree	1225 (33.8)	329 (9.1)	441 (12.2)	1625 (44.9)	
Post Graduate Degree	1318 (35.1)	326 (8.7)	471 (12.6)	1636 (43.6)	
Household Income; M = 519					
<$74,999	1579(39.6)	422 (10.6)	512 (12.8)	1477 (37.0)	<0.0001 **
$75,000–$99,999	523 (34.1)	136 (8.9)	203 (13.3)	670 (43.7)	
≥$100,000	1877 (32.3)	495 (8.5)	701 (12.1)	2,737 (47.1)	
PCL ≥ 44; M = 584					
Yes	1389 (48.9)	306 (10.8)	345 (12.2)	799 (28.1)	<0.0001 ***
No	2504 (29.7)	734 (8.7)	1073 (12.7)	4117 (48.9)	
Sources of Social Integration; M = 221					
0	52 (44.4)	9 (7.7)	22 (18.8)	34 (29.1)	<0.0001 **
1	281 (47.7)	56 (9.5)	60 (10.2)	192 (32.6)	
2	1799 (34.5)	482 (9.2)	619 (11.9)	2321 (44.5)	
3	1387 (35.2)	355 (9.0)	502 (12.8)	1692 (43.0)	
4	536 (30.3)	173 (9.8)	266 (15.1)	792 (44.8)	
Social Support					
Mean	12.3	12.7	13.1	13.8	<0.0001 **
SD	5.4	5.4	5.4	5.2	
PHQ-8 Score					
Mean	9	7.6	6.8	5.4	<0.0001 **
SD	6.7	6.1	5.9	5.3	

Abbreviation: mths: months; N: number; M: missing; SD: standard deviation; PCL: PTSD Checklist; PHQ-8: Personal Health Questionnaire Depression Scale 8th Edition. * row percentages; ** *p*-Value using bivariate multinomial regression; *** *p*-Value using chi-square test.

**Table 3 ijerph-19-07113-t003:** Sociodemographic and Mental Health Characteristics of World Trade Center Health Registry Enrollees who had at Least one Session of Counselling or Therapy After 9/11, Stratified by Perceived Unmet Mental Healthcare Need (i.e., Outcome #2), 2015 to 2016 (*n* = 11,851).

	Perceived Unmet Mental Healthcare Need (Outcome 2)		
Variable	Yes	No	*p*-Value
	*n* (%) *	*n* (%) *	
Totals	1455 (12.3)	10,369 (87.7)	n/a
General Self Efficacy Score; M = 130			
Mean	13.9	15.6	<0.0001 **
SD	3.3	3.1	
**Sex**			
Male	677 (10.6)	5703 (89.4)	<0.0001 ***
Female	778 (14.2)	4693 (85.8)	
Age; M = 3			
18–40 years	225 (14.1)	1371 (85.9)	0.003 ***
35–41 years	1051 (12.5)	7370 (87.5)	
65+ years	178 (9.7)	1653 (90.3)	
Education; M = 88			
Highschool/GED or less	178 (14.2)	1076 (85.8	<0.0001 ***
At least some college or technical degree	445 (14.2)	2693 (85.8)	
Bachelor’s Degree	434 (12.0)	3186 (88.0)	
Post Graduate Degree	388 (10.3)	3363 (89.7)	
Household Income; M = 519			
<$74,999	698 (17.5)	3292 (82.5)	<0.0001 ***
$75,000–$99,999	181 (11.8)	1351 (88.2)	
≥$100,000	519 (8.9)	5291 (91.1)	
PCL ≥ 44; M = 584			
Yes	642 (22.6)	2197 (77.4)	<0.0001 ***
No	719 (8.5)	7709 (91.5)	
Sources of Social Integration; M = 221			
0	40 (34.2)	77 (65.8)	<0.0001 ***
1	111 (18.9)	478 (81.2)	
2	735 (14.1)	4486 (85.9)	
3	381 (9.7)	3555 (90.3)	
4	150 (8.5)	1617 (91.5)	
Social Support; M = 241			<0.0001 **
Mean	10.4	13.5	
SD	5.5	5.2	
PHQ-8 Score; M = 499			
Mean	11.2	6.4	<0.0001 **
SD	6.3	5.9	

Abbreviation: N: number; M: missing; SD: standard deviation; PCL: PTSD Checklist; PHQ-8: Personal Health Questionnaire Depression Scale 8th Edition. * row percentages; ** *p*-Value using bivariate regression; *** *p*-Value using chi-square test.

**Table 4 ijerph-19-07113-t004:** Adjusted Odds Ratios (AORs) of Outcomes 1–3 (1. Most recent counselling or therapy visit, 2. Perceived unmet mental healthcare need, 3. Perception of helpfulness of the most recent mental health visit) and Self Efficacy Score), 2015 to 2016, (*n* = 5245).

Variable	Self Efficacy Score: AOR	95% CI (Low)	95% CI (High)
Most Recent Counselling or Therapy Treatment Visit			
<4 months ago	REF	REF	REF
4 month–less than 1 year ago	1.06	1.03	1.09
1–2 years ago	1.07	1.04	1.12
2+ years ago	1.10	1.08	1.12
Perceived Unmet Mental Healthcare Need			
Yes	REF	REF	REF
No	0.99	0.98	1.02
Perception of Helpfulness of Most Recent Mental Health Treatment Visit			
Not at all helpful	0.947	0.894	1.002
Slightly helpful	0.915	0.884	0.947
Somewhat helpful	0.932	0.907	0.958
Very helpful	REF	REF	REF

Note: all models were adjusted for age, sex, income, education, PTSD, PHQ-8 Score, Sources of Social Integration, and Social Support. Abbreviations: AOR: Adjusted odds ratio, CI: confidence interval.

**Table 5 ijerph-19-07113-t005:** Sociodemographic and Mental Health Characteristics of World Trade Center Health Registry Enrollees Who Had at Least One Session of Counselling or Therapy after 9/11, Stratified by Perception of Helpfulness of last Counseling or Therapy Appointment (i.e., Outcome #3), 2015 to 2016 (*n* = 5245).

		Perception of Helpfulness of Most Recent Counselling or Therapy Visit (Outocome 3)				
	Total	Very helpful	Somewhat helpful	Slightly helpful	Not at all helpful	*p*-Value
Variable	*n* (%) *	*n* (%) *	*n* (%) *	*n* (%) *	*n* (%) *	
Totals	5162 (100)	2145 (41.6)	1969 (38.1)	846 (16.4)	202 (3.9)	n/a
General Self Efficacy Score						
Mean	14.6	15.5	14.4	13.4	12.8	<0.0001 **
SD	3.4	3.2	3.2	3.6	4.2	
**Sex**						
Male	2795 (54.1)	1019 (36.5)	1134 (40.6)	519 (18.6)	123 (4.4)	<0.0001 ***
Female	2367 (45.9)	1126 (47.6)	835 (35.3)	327(13.8)	79 (3.3)	
Age						
18–40 years	703 (13.7)	301 (42.8)	259 (36.8)	110 (15.7)	33 (4.7)	0.0516 ***
41–64 years	3664 (71.0)	1479 (40.4)	1417 (38.7)	628 (17.1)	140 (3.8)	
65+ years	794 (15.3)	364 (45.8)	293 (36.9)	108 (13.6)	29 (3.7)	
Education; M = 88						
Highschool/GED or less	612 (11.9)	214 (35.0)	232 (37.9)	118 (19.3)	48 (7.8)	<0.0001 ***
At least some college or technical degree	1350 (26.4)	502 (37.2)	519 (38.4)	264 (19.6)	65 (4.8)	
Bachelor’s Degree	1540 (31.1)	663 (43.1)	584 (37.9)	243 (15.8)	50 (3.2)	
Postgrad Degree	1620 (31.6)	750 (46.3)	616 (38.0)	218 (13.5)	36 (2.2)	
Household Income; M = 519						
<$74,999	1962 (39.6)	801 (40.8)	707 (36.1)	352 (17.9)	102 (5.2)	<0.0001 ***
$75,000–$99,999	649 (13.1)	255 (39.3)	240 (37.0)	135 (20.8)	19 (2.9)	
≥$100,000	2343 (47.3)	1007 (43.0)	938 (40.0)	324 (13.8)	74 (3.2)	
PCL ≥44 M: 299						
Yes	1671 (34.6)	511 (30.6)	670 (40.0)	382 (22.9)	108 (6.5)	<0.0001 ***
No	3192 (65.4)	1501 (47.0)	1196 (37.5)	413 (12.9)	82 (2.6)	
Sources of Social Integration						
0	60 (1.2)	11 (18.3)	19 (31.7)	17 (28.3)	13 (21.7)	<0.0001 ***
1	329 (6.5)	83 (25.2)	123 (37.4)	95 (28.9)	28 (8.5)	
2	2248 (44.5)	879 (39.1)	870 (38.7)	405 (18.0)	94 (4.2)	
3	1712 (33.9)	776 (45.3)	646 (37.7)	234 (13.7)	56 (3.3)	
4	704 (13.9)	355 (50.4)	268 (38.1)	72 (10.2)	9 (1.3)	
Social Support						
Mean	12.5	13.5	12.2	10.7	10.9	<0.0001 **
SD	5.4	5.2	5.3	5.4	5.8	
PHQ-8 Score						
Mean	8.7	6.7	9.1	11.5	13.5	<0.0001 **
SD	6.6	5.6	6.2	6.9	7.5	

Abbreviation: n: number; M: missing; SD: standard deviation; GED: General Equivalency Diploma; PCL: PTSD Checklist; PHQ-8: Personal Health Questionnaire Depression Scale 8th Edition. * row percents; ** *p*-Value using bivariate multinomial regression; *** *p*-Value using chi-square test.

## Data Availability

The data presented in this study are available on request from the World Trade Center Registry. The data are not publicly available due to privacy reasons.

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
