# Peer review of "Self-Efficacy and Mental Health Help-Seeking Behavior of World Trade Center Health Registry Enrollees, 2015–2016"

_ijerph, 2022, doi:10.3390/ijerph19127113_

Round 1

Reviewer 1 Report

Dear Authors

In general, the manuscript makes a complete and sufficient review of the antecedents related to self-efficacy and help-seeking in mental health. In the introduction and problematization I have no particular observations, I believe that it is sufficiently addressed and the research problem is properly positioned, however, I have some doubts that I place below:

  1. Data analysis and results

Why was it decided to use the chi-square test for the analysis of dichotomous variables? The chi-square test is sensitive to sample size, that is, in large samples, such as the one in this study, it tends to become significant, increasing the possibility of incurring in type 1 errors. If possible, it is suggested to the authors that, instead of chi-square, use a student's t-test, in order to avoid threats to the validity of the statistical conclusion.

Regarding the use of ANOVA, the suggestion is that the homoscedasticity test is also included in order to verify that the variances are equal between the different levels of the factor, for example, in the case of the variable "education" in each one of its four levels, the variances would be expected to be homogeneous, however, if they are heterogeneous, a robust test should be chosen (games – howell, for example). Additionally, it would be convenient to place the contrasts between the different levels of the factor, since the ANOVA only indicates if there are differences at some level, however, it does not specify which one, for this the post-hoc tests must be shown.

If the authors choose to perform the analyzes of the dichotomous variables with Student's t, the equality of variances should also be verified.

In relation to multinomial logistic regression, it is suggested that the authors present the evidence in relation to the evaluation of multicollinearity between the independent variables, since this analysis assumes that there should not be high and significant correlations, so an analysis of correlations between the independent variables is enough to show that there are no collinearity problems.

Likewise, at the end of the data analysis section, it is suggested to place some examples of the confounding factors that they controlled for and how they statistically controlled for them, although this is mentioned as a note in the results tables, it is considered appropriate to place them in the method section as well.

It is worth noting that the authors did an excellent job in the careful selection of the work sample, so I believe that with these adjustments the methodological part and the presentation of results can be improved.

Finally, I consider that the discussion and the conclusions are pertinent according to the objectives of the study and I completely agree with the authors that, according to the sample and type of study, it is not entirely possible to establish the dependency-independence relationships between the variables involved, however, this does not prevent visualizing that self-efficacy has an important role in seeking mental health care.

Congrats for this excellent work!

Author Response

Response to Reviewer 1 Comments

Point 1: In general, the manuscript makes a complete and sufficient review of the antecedents related to self-efficacy and help-seeking in mental health. In the introduction and problematization I have no particular observations, I believe that it is sufficiently addressed and the research problem is properly positioned, however, I have some doubts that I place below:

Response 1: we thank you very much for taking the time to reviewer our paper and for allowing us to address your concerns. We have addressed and responded to each point you made in your review of our manuscript, which we discuss in this response document.

Point 2: Why was it decided to use the chi-square test for the analysis of dichotomous variables? The chi-square test is sensitive to sample size, that is, in large samples, such as the one in this study, it tends to become significant, increasing the possibility of incurring in type 1 errors. If possible, it is suggested to the authors that, instead of chi-square, use a student's t-test, in order to avoid threats to the validity of the statistical conclusion.

Response 2: Thank you very much for raising this issue. We see now how our tables made the distinction between our outcomes and exposure variables confusing. In our study, we have three outcome variables, all of which are categorical: 1) time of most recent counselling or therapy visit (4 levels); 2) Perceived unmet healthcare need (dichotomous); 3) Perception of helpfulness of most recent mental health treatment visit (4 levels). Our exposure, self-efficacy score, is continuous, but, again, out three outcomes are categorical. That is why we performed chi-square tests for independence between our dichotomous covariates and categorical/dichotomous outcomes. We have changed Tables 1 through 3 and Table 5 to make this more clear. We have also corrected the language on lines 203 and 204. They now say, “We used Chi square tests for dichotomous variables bivariate logistic regression to tests for continuous variables to assess their association with the outcomes.”

Point 3: Regarding the use of ANOVA, the suggestion is that the homoscedasticity test is also included in order to verify that the variances are equal between the different levels of the factor, for example, in the case of the variable "education" in each one of its four levels, the variances would be expected to be homogeneous, however, if they are heterogeneous, a robust test should be chosen (games – howell, for example). Additionally, it would be convenient to place the contrasts between the different levels of the factor, since the ANOVA only indicates if there are differences at some level, however, it does not specify which one, for this the post-hoc tests must be shown.

Response 3: Thank you for this feedback. We no longer use ANOVA in our analyses. Instead, we have used bivariate multinomial regression to test for significance between all three of our categorical outcomes and our three continuous explanatory variables: self-efficacy, PHQ-8 score, and social support. Please see response to point 2 for greater detail.

Point 4: If the authors choose to perform the analyzes of the dichotomous variables with Student's t, the equality of variances should also be verified.

Response 4: We have not performed a student’s t-test because our three outcome variables are categorical. We have altered our tables to make this clearer.  Please see our response to Point 2 for more detail.

Point 5: In relation to multinomial logistic regression, it is suggested that the authors present the evidence in relation to the evaluation of multicollinearity between the independent variables, since this analysis assumes that there should not be high and significant correlations, so an analysis of correlations between the independent variables is enough to show that there are no collinearity problems.

Response 5: Thank you for bringing this to out attention. In response to your suggestion, we tested for multicollinearity between all our study’s predictor variables by way of a correlation matrix. Using “proc corr” in SAS, we examined the pairwise correlation of each of our study variables, looking for any correlation coefficients that exceeded the cut-off point of 0.8, recommended by Berry & Feldman (1985). All of our pair-wise correlations were well below the 0.8 cutoff, and we were able to be concluded that multicollinearity was not detected between any of the predictor variables in this study.

Source used:

Berry, W.D., Feldman, S. Multiple Regression in Practice (Quantitative Applications in the Social Sciences). SAGE Publications; Thousand Oaks. CA: 1985.

Point 6: Likewise, at the end of the data analysis section, it is suggested to place some examples of the confounding factors that they controlled for and how they statistically controlled for them, although this is mentioned as a note in the results tables, it is considered appropriate to place them in the method section as well.

Response 6: Thank you for this suggestion. We agree that the way has originally included the information on the confounding variables in the body of the paper was not clear. To address this, we reformatted the “Confounders” subheading on lines 155 – 156 to make the demarcation of this section more obvious. Further, in the “Data Analysis” section of the paper, we included the names of all confounding variables used in the analysis, in parentheses on lines 215 – 216

Point 7: It is worth noting that the authors did an excellent job in the careful selection of the work sample, so I believe that with these adjustments the methodological part and the presentation of results can be improved.

Response 7: We thank you for your kind words and have done our best to address all your suggested adjustments to the methods and results.

Point 8:  Finally, I consider that the discussion and the conclusions are pertinent according to the objectives of the study and I completely agree with the authors that, according to the sample and type of study, it is not entirely possible to establish the dependency-independence relationships between the variables involved, however, this does not prevent visualizing that self-efficacy has an important role in seeking mental health care. Congrats for this excellent work!

Response 8: Thank you, again, for taking the time to review our manuscript and for providing your thoughtful feedback.

Reviewer 2 Report

This article addresses an interesting topic, and it is also very well written overall. I will recommend a minor revision. My mainly concern is the discussion which I find difficult to follow. There are also links made with very different subjects, which I am not sure are very pertinent. I will recommend revising and reducing the discussion, and more focusing it on the key results and recommendations towards better care.  In the limitations, the fact that the team did not use the other waves of data collected, which were possible with this database, should be added.

Author Response

Response to Reviewer 2 Comments

Point 1: This article addresses an interesting topic, and it is also very well written overall. I will recommend a minor revision.

Response 1: We thank you for the opportunity to address your concerns with our manuscript. Thank you for taking the time to review it.

Point 2:  My mainly concern is the discussion which I find difficult to follow.

Response 2: Thank you for your feedback on our discussion. We have synthesized it further in this iteration and we hope that it clearer because of our edits throughout.

Point 3: There are also links made with very different subjects, which I am not sure are very pertinent.

Response 3: Thank you for highlighting this. We have removed the link we made to breast-feeding self-efficacy in our discussion, which we felt, upon further consideration, was too far a logical leap in our reasoning.

Point 4: I will recommend revising and reducing the discussion, and more focusing it on the key results and recommendations towards better care

Response 4: Thank you very much for this insight. We have gone through our discussion synthesized as much as possible to highlight our main results and recommendations.

Point 5: In the limitations, the fact that the team did not use the other waves of data collected, which were possible with this database, should be added.

Response 5: Please see lines 381-383 in the manuscript, where we included a sentence on the limitation you have specified in this point.